# Statins use and COVID-19 outcomes in hospitalized patients

**Samuel K. Ayeh[1], Enoch J. Abbey[1], Banda A. A. Khalifa[2], Richard D. Nudotor[3], Albert Danso Osei[4], Vignesh Chidambaram[1], Ngozi Osuji[1], Samiha Khan[1], Emmanuella L. Salia[5], Modupe O. Oduwole[1], Hasiya E. Yusuf[6], Oluwatobi Lasisi[7], Esosa Nosakhare[8], Petros C. Karakousis[1,9]***

1 Department of Medicine, Johns Hopkins School of Medicine, Baltimore, MD, United States of America, 2 Department of Epidemiology, Johns Hopkins Bloomberg School of Public Health, Baltimore, MD, United States of America, 3 Department of Surgery, Johns Hopkins School of Medicine, Baltimore, MD, United States of America, 4 Department of Internal Medicine, Medstar Union Memorial Hospital, Baltimore, MD, United States of America, 5 Department of Genetics, Genomics and Informatics, University of Tennessee Health Science Center, Memphis, TN, United States of America, 6 Department of Pediatrics, Johns Hopkins School of Medicine, Baltimore, MD, United States of America, 7 Department of Internal Medicine, Wayne State University School of Medicine, Detroit, MI, United States of America, 8 Armstrong Institute for Patient Safety and Quality, Johns Hopkins Medicine, Baltimore, MD, United States of America, 9 Department of International Health, Johns Hopkins Bloomberg School of Public Health, Baltimore, MD, United States of America

* petros@jhmi.edu

## Abstract

### Background

There is an urgent need for novel therapeutic strategies for reversing COVID-19-related lung inflammation. Recent evidence has demonstrated that the cholesterol-lowering agents, statins, are associated with reduced mortality in patients with various respiratory infections. We sought to investigate the relationship between statin use and COVID-19 disease severity in hospitalized patients.

### Methods

A retrospective analysis of COVID-19 patients admitted to the Johns Hopkins Medical Institutions between March 1, 2020 and June 30, 2020 was performed. The outcomes of interest were mortality and severe COVID-19 infection, as defined by prolonged hospital stay ($\geq$ 7 days) and/ or invasive mechanical ventilation. Logistic regression, Cox proportional hazards regression and propensity score matching were used to obtain both univariable and multivariable associations between covariates and outcomes in addition to the average treatment effect of statin use.

### Results

Of the 4,447 patients who met our inclusion criteria, 594 (13.4%) patients were exposed to statins on admission, of which 340 (57.2%) were male. The mean age was higher in statin users compared to non-users [64.9 ± 13.4 vs. 45.5 ± 16.6 years, p <0.001]. The average treatment effect of statin use on COVID-19-related mortality was RR = 1.00 (95% CI: 0.99–

**Data Availability Statement:** Data cannot be shared publicly because it is a hospital-based data. Data are available from the Johns Hopkins Medicine Institutional Data Access / Ethics Committee (contact via ictr@jhmi.edu) for

researchers who meet the criteria for access to confidential data.

**Funding:** This retrospective cohort study was supported by the National Institute of Allergy and Infectious Diseases (NIAID)/ National Institutes of Health (NIH) grants UH3AI122309 and K24AI143447 to P.C.K. The funders had no role in study design, data collection and analysis, decision to publish, or preparation of the manuscript.

**Competing interests:** The authors have declared that no competing interests exist.

1.01, p = 0.928), while its effect on severe COVID-19 infection was RR = 1.18 (95% CI: 1.11–1.27, p <0.001).

## Conclusion

Statin use was not associated with altered mortality, but with an 18% increased risk of severe COVID-19 infection.

## 1. Background

A novel beta-coronavirus, now known as SARS-CoV-2, was first reported to cause severe pneumonia in China's Wuhan province in late December 2019 [1]. Despite early efforts to contain the virus, coronavirus disease 2019 (COVID-19) has been reported in almost all other countries, attaining pandemic status by March 2020 [1].

Although vaccines against COVID-19 have shown promising results, treatment modalities such as remdesivir, dexamethasone, and the monoclonal antibody treatment REGEN-COV2 continue to play essential roles in treating high-risk infected patients or those with moderate to severe disease [2]. The medical community has continued to look for inexpensive, practical, and rapidly deployable agents to improve mortality and reduce the need for mechanical ventilation among hospitalized patients. Recently, 3-hydroxy-3-methyl-glutaryl-CoA (HMG-CoA) reductase inhibitors (statins) have gained attention for their potential utility as adjunctive, host-directed therapies against various infectious diseases [3]. In preclinical tuberculosis (TB) models, adjunctive statin therapy enhanced the bactericidal activity of the first-line antitubercular regimen [4, 5] and shortened the duration of curative TB treatment [6]. Similarly, statins have also been shown to potentiate host antiviral responses in preclinical models [7]. For example, statins inhibit HIV replication in CD4$^+$ T cells and suppress cytomegalovirus and respiratory syncytial virus loads in mice [8–10]. When added as a vaccine adjuvant, simvastatin enhanced protection relative to influenza A/PR8 hemagglutinin A1 alone against influenza challenge in mice and macaques [11]. Apart from inhibiting the first step of cholesterol biosynthesis, statins possess anti-inflammatory and immune-modulating functions [12, 13]. These effects have been attributed to their modulation of various host molecular pathways, including inhibition of activation of the mechanistic target of rapamycin complex 1 (mTORC1), activation of AMP-activated protein kinase (AMPK), enhanced nuclear translocation of transcription factor EB (TFEB), modulation of NF-κB activity, as well as inhibition of protein prenylation [14–17].

A randomized control trial found that statins, when added to the standard treatment, reduced mortality in elderly patients with community-acquired pneumonia and sepsis [18]. Several retrospective studies have concluded that statin use is associated with reduced mortality in patients with bacteremia, sepsis, and pneumonia [19]. The use of statins was also associated with lower death rates and intubation in patients admitted with viral pneumonia [20]. More recently, a retrospective study in an elderly care facility reported that statin use in residents diagnosed with COVID-19 was associated with less severe symptoms and improved clinical outcomes [21]. Several studies have suggested that statin use is also associated with reduced COVID-19-related mortality [22–24]. However, prior work showed that expression of the ACE2 receptor, which is required for SARS-CoV-2 entry [25] into airway epithelial cells, is upregulated by statin therapy in various tissues [26, 27], raising theoretical concerns about the use of statins in the context of COVID-19 infection.

In this retrospective study, we sought to investigate the relationship between statin use and COVID-19 mortality and disease severity in patients admitted to the Johns Hopkins health system.

## 2. Materials and methods

### 2.1. Study design and participants

This is a retrospective, multi-center cohort study of 4,447 patients hospitalized at the Johns Hopkins Hospital and affiliated hospitals (Johns Hopkins Bayview Medical Center, Howard County General Hospital, Sibley Memorial Hospital, and Suburban Hospital) with a diagnosis of SARS-CoV-2 infection between March 1, 2020 and June 30, 2020. The study inclusion criteria included: 1) inpatients diagnosed with COVID-19; 2) age ≥ 18 years old. There were no specific exclusion criteria. The data source was the Johns Hopkins Crown database, which contains comprehensive information on study participants, including detailed demographics, past medical history, medications, and hospital course. The study was approved by the Johns Hopkins Institutional Review Board and the Johns Hopkins Covid-19 and Data Research Evaluation (CADRE) Committees.

### 2.2. Exposure and outcomes

The primary endpoints were mortality (in-hospital death) and severe COVID-19 infection, defined as prolonged hospital stay (≥ 7 days) and/or need for invasive mechanical ventilation. The main exposure of interest was statin use, based on the medication administration record. Follow-up in this study was defined as the period between hospital admission and the occurrence of an outcome, until discharge, or until the time of administrative censoring (June 30, 2020), whichever occurred first. Demographic data included age, sex, race, and employment status. Relevant clinical data shown to be associated with mortality and poor outcomes in prior studies [28], such as body mass index (BMI), Elixhauser comorbidity index, history of hypertension (≥140 mm Hg systolic / ≥ 90 mm Hg diastolic), diabetes mellitus, chronic kidney disease patients on interval dialysis, use of angiotensin-converting enzyme (ACE) inhibitors or angiotensin II receptor blockers (ARBs), and serum Interleukins-6 (IL-6) levels, were extracted from the database. BMI was categorized as normal (18.5–24.9 kg/m$^2$), overweight (25.0–29.9 kg/m$^2$) and obese (≥ 30.0kg/m$^2$). Generic names of statins and angiotensin-converting enzyme (ACE) inhibitors/ARB prescribed are listed in **S1 and S2 Tables.** Serum IL-6 was categorized as a binary variable, i.e., either high (>3.2 pg/ml) or normal (≤ 3.1 pg/ml). Employment status was divided into the following categories: employed, unemployed, student, and retired/disabled. The AHRQ-Elixhauser comorbidity index, designed to assess the risk of in-hospital mortality and 30-day readmission, was used [29], with each condition assigned an index weight.

### 2.3. Statistical analyses

Cohort baseline and time-specific characteristics were compared by statin treatment status, as indicated in **Table 1**. Single measure baseline comparisons were made using Chi-square ($\chi^2$) tests for categorical variables and one-way ANOVA or Kruskal-Wallis tests for continuous variables.

The association (risk factor analyses) between covariates and primary outcomes (mortality and severe COVID-19 infection) was evaluated using Cox proportional hazards regression and logistic regression to obtain both univariable and multivariable regression models and point estimates (**Tables 2 and 3**). Variables used in the regression models were based on both

**Table 1. Baseline characteristics of statin users versus statin non-users.**

| Characteristics | Statin use | No statin use | p-value |
|---|---|---|---|
| | (n = 594) | (n = 3853) | |
| **Age, years Mean (SD)** | 64.9 (13.4) * | 45.5 (16.6) * | <0.001 |
| **Male–No. (%)** | 340 (57.2) | 1882 (48.9) | <0.001 |
| **Race** | | | <0.001 |
| *Black* | 257 (46.6) | 1248 (34.5) | |
| *Other* | 105 (19.0) | 1553 (42.9) | |
| **BMI (kg/m²)** | | | <0.001 |
| *Overweight (25.0–29.9)* | 133 (23.2) | 705 (18.6) | |
| *Obese (≥30.0)* | 337 (58.8) | 2,609 (68.8) | |
| **Hypertension** | 439 (80.3) | 973 (28.4) | <0.001 |
| **Diabetes** | 317 (57.9) | 503 (14.7) | <0.001 |
| **ACE inhibitor /ARB use** | 214 (36.0) | 267 (9.8) | <0.001 |
| **Interval Dialysis** | 41 (6.9) | 28 (0.7) | <0.001 |
| **IL-6 (High)** | 9 (1.5) | 56 (1.5) | 0.91 |
| **Employment Status** | | | |
| *Unemployed* | 140 (23.6) | 1,294 (34.1) | <0.001 |
| *Students* | 1 (0.2) | 56 (1.5) | |
| *Retired/ Disabled* | 318 (53.6) | 612 (16.1) | |
| **Total Elixhauser Score** | 7.3 (11.3) * | 1.7 (7.6) * | <0.001 |

N/n: Number of patients, %: percent of patients, CI: 95% confidence interval, BMI: Body mass index, Kg/m², */SD: standard deviation, ACE/ARB: Angiotensin converting enzyme inhibitor or Aldosterone receptor blocker. Significant p-values <0.05.

a conceptual framework and stepwise regression using a p-value of <0.05. Backward stepwise regression models were conducted after a model had been designed based on a conceptual framework. The final model for mortality contained the following variables: age, sex, race,

**Table 2. Cox proportional regression showing association between covariables and mortality.**

| Covariable | Unadjusted Hazard Ratio (95% CI) | Adjusted Hazard Ratio (95% CI) |
|---|---|---|
| **Age (years)** | 1.04 (1.02–1.07) | 1.04 (1.01–1.06) |
| **Male** | 1.48 (0.87–2.52) | 1.78 (1.01–3.17) |
| **Race** | | |
| *Black* | 0.70 (0.40–1.24) | 0.97 (0.53–1.76) |
| *other* | 0.44 (0.20–0.93) | 0.78 (0.34–1.81) |
| **BMI** | | |
| *Overweight* | 0.56 (0.27–1.15) | - |
| *Obese* | 0.57 (0.31–1.04) | |
| **Hypertension** | 1.39 (0.81–2.38) | - |
| **Diabetes** | 1.96 (1.18–3.28) | - |
| **ACE/ ARB Use** | 0.69 (0.40–1.18) | - |
| **Interval Dialysis** | 1.02 (0.41–2.56) | - |
| **IL-6 (High)** | - | - |
| **Employment Status** | | |
| *Unemployed* | 1.08 (0.44–2.67) | - |
| *Students* | - | |
| *Retired/ Disabled* | 3.75 (1.88–7.49) | |
| **Total Elixhauser Score** | 1.05 (1.03–1.07) | 1.03 (1.01–1.05) |

**Table 3. Logistic regression showing association between covariables and severe COVID-19 infection.**

| Covariable | Unadjusted Relative Risk | Adjusted Relative Risk |
|---|---|---|
| **Age (years)** | 1.04 (1.03–1.04) | 1.80 (1.38–2.35) |
| **Male** | 1.48 (1.24–1.76) | 1.50 (1.21–1.85) |
| **Race** | | - |
| *Black* | 0.86 (0.69–1.07) | |
| *Other* | 0.53 (0.42–0.66) | |
| **BMI** | | - |
| *Overweight* | 1.13 (0.85–1.50) | |
| *Obese* | 0.67 (0.52–0.86) | |
| **Hypertension** | 2.95 (2.46–3.54) | 0.74 (0.57–0.97) |
| **Diabetes** | 3.48 (2.87–4.20) | 1.66 (1.30–2.12) |
| **ACE/ ARB Use** | 10.07 (8.13–12.48) | 7.18 (5.64–9.14) |
| **Interval Dialysis** | 11.56 (7.05–18.96) | 3.49 (1.94–6.28) |
| **IL-6 (High)** | 3.19 (1.88–5.41) | 2.85 (1.55–5.24) |
| **Employment Status** | | - |
| *Unemployed* | 1.22 (0.98–1.52) | |
| *Students* | 0.52 (0.16–1.67) | |
| *Retired/ Disabled* | 3.07 (2.49–3.79) | |
| **Total Elixhauser Score** | 1.03 (1.03–1.04) | - |

statin-use, and Elixhauser comorbidity index; while that of severe COVID-19 infection included the following variables: age, sex, IL-6 levels, statin-use, ACE inhibitor/ ARB use, hypertension, diabetes mellitus, and chronic kidney disease. Model selection for propensity score matching (PSM) involved backward stepwise regression, Akaike information criterion, and model fit determined using Hosmer-Lemeshow test. PSM involved a 1:1 nearest neighbor matching with replacement to minimize conditional bias. The balance of the propensity model between the treated and control groups was assessed by comparing the raw and matched standard differences and variance ratios. A well-balanced matching model produces a standard difference value close to 0 and a variance ratio close to 1 for each covariate, as well as a balanced boxplot (**S3 and S4 Tables** and **S1 and S2 Figs**). All statistical analyses were performed using STATA® (Statistical Data Analysis Package version 16.0 IC, College Station, TX–USA).

## 3. Results

### 3.1. Patient characteristics

A total of 4,447 inpatients with COVID-19 met study eligibility criteria, with 594 (13.4%) receiving statins upon admission (**Table 1**). Statin users were significantly more likely to be male (57.2% vs. 48.9%, p <0.001) and older (64.9 ± 13.4 vs. 45.5 ± 16.6, p <0.001) compared to statin non-users. The largest percentage of statin users were Black (46.6%) and had hypertension (73.9%) and diabetes (53.4%). Statin users were more likely to be receiving ACE inhibitors/ARBs compared to non-statin users and to experience more frequent episodes of interval dialysis, possibly due to more profound disease and/or the presence of more comorbidities. Most statin users had higher Elixhauser comorbidity scores than non-users (7.3 ± 11.3 vs 1.7 ± 7.6, p <0.001).

### 3.2. Risk factors for COVID-19-related mortality

The association between patient characteristics and in-hospital mortality was evaluated via univariable and multivariable analysis using Cox proportional regression (**Table 3**). Based on

**Table 4. Risk for mortality and severe COVID-19 infection in statin users versus statin non-users.**

| Outcome | Crude Risk | MV-adjusted Risk | PSM-Relative Risk |
|---|---|---|---|
| **Mortality** | | | |
| Statin versus non-statin | HR = 1.16 (0.70–1.92) | HR = 0.92 (0.53–1.59) | RR = 1.00 (0.99–1.01) |
| **Severe COVID-19** | | | |
| Statin versus non-statin | RR = 6.17 (5.07–7.52) | RR = 1.80 (1.38–2.35) | RR = 1.18 (1.11–1.27) |

the univariable analysis, every yearly increase in age was associated with a 4% increased hazard of death among inpatients [HR = 1.04, 95% CI (1.02–1.07)] and this was similar in the adjusted model [HR = 1.04, 95% CI (1.01–1.06)]. Although statin use was associated with an increased hazard of death in the unadjusted model, the adjusted model (Table 4) showed a protective effect of HR = 0.92, 95% CI (0.53–1.59). However, this effect was not statistically significant. On the other hand, male patients with COVID-19 experienced increased mortality in both the unadjusted (HR = 1.48, 95% CI [0.87–2.52]) and adjusted (HR = 1.78, 95% CI [1.01–3.17]) models. Increasing Elixhauser scores were associated with an increased hazard of death, both in univariable and multivariable analyses [HR = 1.05, 95% CI (1.03–1.07); multivariable-adjusted HR = 1.03, [95% CI (1.01–1.05)] (Table 4).

The propensity-score matching analysis created 2,789 matches in the statin use (exposure) and statin non-use (control) groups. All assessed variables were balanced between treatment and control groups, as demonstrated by the standardized differences and variance ratios provided in S3 Table. The average treatment effect of statin use on COVID-19-related mortality in the matched groups was not statistically significant (RR = 1.00; 95% CI [0.99–1.01]).

### 3.3. Risk factors for severe COVID-19 infection

Increasing age and male sex were both associated with an increased risk of severe COVID-19 infection by univariable and multivariable analysis (Table 3). Similarly, the presence of diabetes and elevated IL-6 levels, and exposure to interval dialysis were associated with an increased risk of severe COVID-19 infection among inpatients by univariable and multivariable analyses. Also, the risk for severe COVID-19 infection was higher with the use of ACE inhibitors or ARBs (Table 3, MV-adjusted RR = 7.18 95% CI [5.64–9.14]), as well as statin use (Table 4, MV-adjusted, RR = 1.80; 95% CI [1.38–2.35], even after adjusting for other relevant covariables. The propensity-score matching analysis created 2,789 matches in the statin group and the statin non-use group. All assessed variables were balanced between the two groups, as demonstrated by the standardized differences and variance ratios provided in S4 Table. Statin use was associated with an 18% increased risk of severe disease among hospitalized patients (RR = 1.18, 95% CI [1.11–1.27]) (Table 4).

### 4. Discussion

Our study sought to investigate the potential association between statin use and COVID-19 disease severity or mortality. We found no association between statin use and mortality in this cohort, but there was a statistically significant association between statin use and increased COVID-19 disease severity, defined as prolonged hospital stay ($\geq$ 7 days) and/or need for invasive mechanical ventilation. This association was demonstrated by multivariable logistic regression and was sustained in the analysis after propensity score matching.

In a similar retrospective cohort study from Belgium, De Spiegeleer et al. found statin use to be independently associated with the absence of symptoms during COVID-19, but not with severe or fatal infection among a limited sample of 154 elderly nursing home residents with

COVID-19 [21]. Another retrospective study of 87 patients admitted to the ICU with COVID-19 linked statin non-use to a 73% higher probability of more rapid progression to mortality after adjusting for age, hypertension, cardiovascular disease, invasive mechanical ventilation, disease severity, and other adjuvant therapies [30]. However, the authors did not obtain consistent results with the chi-square test (p-value = 0.20) between statin use and mortality. Additionally, the sample size was low in the study (< 25 patients receiving statin therapy. A recent meta-analysis of 9 observational studies using unadjusted data showed that statin use was not associated with increased mortality or severe disease [31].

Our study findings are in contrast to those of Daniels *et al.*, who reported that statin use in the month prior to hospitalization was associated with a 71% percent risk reduction in severe COVID-19 disease [22] and those of Rossi *et al.*, who reported a non-significant reduction in mortality with statin use [23]. A similar finding of 24% reduction in COVID-19-related mortality among statin users was reported by Peymani *et al.* [24]. In contrast to these reported studies, we found that statin use had no effect on COVID-19-related mortality but increased the risk of COVID-19 disease severity by 18%. Our discrepant findings may be explained by the much smaller sample sizes in the studies by Daniels *et al.* (170 patients with COVID-19; 46 statin users) [22], Rossi *et al.* (71 patients with COVID-19; 42 statin users) [23], and Peymani *et al.* (150 study participants; 75 statin users) [24]. In contrast, our study included a total of 4,447 inpatients with COVID-19, 594 of whom were statin users, allowing us to avoid making a type I error related to statin use and mortality due to small sample size. The endpoints in our study differed from those of the previously published studies. For example, our study defined severe COIVD-19 disease as prolonged hospital stay ($\geq$ 7 days) and/or need for invasive mechanical ventilation, whereas this parameter was defined as death or admission to the intensive care unit by Daniels *et al.* [22]. Alternatively, the divergent findings may be due to differences in the demographic makeup of participants in each study (e.g., elderly individuals or those identifying as Black or Latinx) and/or differences in the prevalence of co-morbidities, such as diabetes mellitus or cardiovascular disease, which are known to be associated with worse COVID-19 clinical outcomes. Our findings are consistent with recently published data highlighting the potential deleterious effects of statins on COVID-19 clinical outcomes, such as mortality within 7 and 28 days of hospital admission and the need for tracheal intubation in patients with type 2 diabetes [32].

Plausible mechanisms of statin-induced exacerbation of COVID-19 infection includes activation of Toll-like receptors and signaling through myeloid differentiation primary response 88 (MyD88) and NF-κB, thereby increasing lung inflammation [33, 34]. As in the case of sepsis, community-acquired pneumonia and TB [35–37], hypolipidemia in patients with COVID-19 has been associated with increased disease severity, and inflammatory markers, such as C-reactive protein and IL-6 were inversely correlated with serum total cholesterol and low-density lipoprotein (LDL) levels [38]. However, it is unclear if statin-mediated reduction in circulating lipid levels is causally linked to more extensive lung inflammation in COVID-19. SARS-CoV-2 reduces the biosynthesis of LDL by altering liver function and increasing vascular permeability, causing a leakage of plasma cholesterol and lipid molecules into alveolar spaces, thereby reducing plasma lipid levels.

Statins have been shown to increase cellular expression of angiotensin-converting enzyme 2 (ACE2) [27], the primary receptor used by SARS-CoV 2 to gain entry into lung cells [39]. Of note, the patient populations most likely to be prescribed statins for their cardioprotective effects, i.e., males, the elderly, and those with hypertension, diabetes, dyslipidemia, and cardiovascular disease, are also the same populations at greatest risk for COVID-19-related mortality [34]. Although generally statins are well tolerated, their use may be associated with rhabdomyolysis and related kidney injury, as well as liver toxicity, which may compound COVID-19

systemic disease [40, 41]. Since most statins are substrates of the hepatic cytochrome P450 system, co-administration of statins with protease inhibitor-based antiretroviral agents or certain immunosuppressive drugs may markedly increase the risks of adverse effects in patients with COVID-19 due to drug interactions [34, 42]. Despite these considerations, continuation of statin therapy is recommended in patients with newly diagnosed COVID-19 infection due to their proven beneficial effects on cardiovascular outcomes in appropriate clinical settings [34].

The heterogeneous nature of our study population and the large sample size, allowed us to assess the relationship between COVID-19 mortality/severity and statin use among a widely varied patient population, thus improving our study's generalizability to other hospitalized patients with COVID-19. The utilization of a backward stepwise regression approach made for a reproducible method for predictor selection in this study, helping to reduce the potential for bias in selecting predictor variables for inclusion in our regression models. Additionally, the use of propensity score matching when comparing statin and non-statin users reduced the potential for selection bias (confounding by indication) and provided a more statistically robust estimate of our primary and secondary outcomes. Furthermore, use of the Elixhauser Score to account for comorbidities in our statistical models provided a more comprehensive approach to account for comorbidities.

Our study has certain limitations. Due to the retrospective study design, it is possible that all potential confounding factors were not properly adjusted. We did not specifically analyze the effects of different types of statins (e.g., lipophilic vs. hydrophilic), as the vast majority of our participants were prescribed atorvastatin, or statin dosing, as this information was not readily available from the database. Furthermore, the possibility of residual confounding and confounding from historical exposure to statins before in-hospital admission/ follow-up could have impacted the association between statin use and the outcomes of interest.

Despite the apparent beneficial effect of statins on outcomes of various infectious diseases, our study revealed that their use was associated with increased COVID-19 disease severity among hospitalized patients. Our findings merit laboratory-based investigations to determine the potential mechanism(s) by which statins may contribute to excessive lung inflammation. Follow-up clinical studies are needed to determine if temporary discontinuation of statins is warranted, particularly among hospitalized patients with COVID-19. Additional studies may also seek to identify if there are specific patient populations with COVID-19 that may benefit from statin therapy.

## Supporting information

**S1 Fig. Balance plot of raw and matched box and whisker plots of statin use versus statin non-use in COVID-19-related mortality.**
(DOCX)

**S2 Fig. Balance plot of raw and matched box and whisker plots of statin use versus statin non-use in severe COVID-19 infections.**
(DOCX)

**S1 Table. Statin medications prescribed at the time of admission.**
(DOCX)

**S2 Table. ACE inhibitors/ARBs prescribed at the time of admission.**
(DOCX)

**S3 Table. Means and variances in raw and balanced data for statin effect on COVID-19-related mortality.**
(DOCX)

**S4 Table. Means and variances in raw and balanced data for statin effect on severe COVID-19 disease.**
(DOCX)

## Author Contributions

**Conceptualization:** Samuel K. Ayeh, Enoch J. Abbey, Banda A. A. Khalifa, Richard D. Nudotor, Emmanuella L. Salia, Modupe O. Oduwole, Hasiya E. Yusuf, Oluwatobi Lasisi, Petros C. Karakousis.

**Formal analysis:** Samuel K. Ayeh, Enoch J. Abbey, Banda A. A. Khalifa, Richard D. Nudotor.

**Funding acquisition:** Petros C. Karakousis.

**Methodology:** Samuel K. Ayeh, Enoch J. Abbey, Banda A. A. Khalifa, Richard D. Nudotor.

**Supervision:** Petros C. Karakousis.

**Writing – original draft:** Samuel K. Ayeh, Enoch J. Abbey, Banda A. A. Khalifa, Richard D. Nudotor, Vignesh Chidambaram, Ngozi Osuji, Samiha Khan, Emmanuella L. Salia, Modupe O. Oduwole, Hasiya E. Yusuf, Oluwatobi Lasisi, Petros C. Karakousis.

**Writing – review & editing:** Samuel K. Ayeh, Enoch J. Abbey, Banda A. A. Khalifa, Richard D. Nudotor, Albert Danso Osei, Vignesh Chidambaram, Ngozi Osuji, Samiha Khan, Emmanuella L. Salia, Modupe O. Oduwole, Hasiya E. Yusuf, Oluwatobi Lasisi, Esosa Nosakhare, Petros C. Karakousis.

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
