## [Decision Letter · Decision Letter 0]

4 Jun 2021

PONE-D-21-15364

Statins use and COVID-19 outcomes in hospitalized patients

PLOS ONE

Dear Dr. Karakousis,

Thank you for submitting your manuscript to PLOS ONE. After careful consideration, we feel that it has merit but does not fully meet PLOS ONE’s publication criteria as it currently stands. Therefore, we invite you to submit a revised version of the manuscript that addresses the points raised during the review process.

We look forward to receiving your revised manuscript.

Kind regards,

Aleksandar R. Zivkovic

Academic Editor

PLOS ONE

Reviewers' comments:

Reviewer #1: The authors have done evaluation from the secondary data regarding the statin use and severe covid-19. My comments are

1. Taking the conclusion in statin use in COVID -19 should be carefull. We should take into account, type, dose statin use, how long the statin have been use before the subject got infection. As we know statin needed time to work the anti inflammmation. should be discussed this issue in the discussion

2. In the results showed that, age, BMI, comorbidity related to the risk of COVID-19. Was the statin use related to severe covid-19 or because any interaction or confounder with comorbidity because they with many co morbidity tend to use statin? should also be discussed.

3. Take the conclusion from the retrospective study should be carefully. May be covid-19 did not improved covid-19 outcome, however further randomized study should confirm this result,

Reviewer #2: First of all, I compliment the authors for their excellent work, which is well-written. In fact, the potential benefit of statins in the COVID-19 scenario is a very important issue, and one that is still controversial.

Below, I respectfully present to the authors my suggestions for improving the manuscript. Please consider the suggestions that you think are relevant.

ABSTRACT

- In Methods: Were patients diagnosed with COVID-19 by laboratory tests? I suggest: "A retrospective analysis of patients with laboratory-confirmed COVID-19 admitted to the Johns Hopkins Medical Institutions between..."

- Please, "risk ratio (RR)".

- In the results, I suggest describing "After propensity-score matching, the average treatment effect..." and/or "after adjusting for age, etc..."

- In Conclusion, I believe that the authors could draw attention to the importance of using PSM, but that, on the other hand, the data should be viewed with caution, given that the study is retrospective.

BACKGROUND

- The first paragraph contains information that has already been published extensively (... firts reported to cause severe pneumonia in China's Wuhan... pandemic

status by March 2020...). Thus, I suggest that the authors start by going more directly to the point, excluding the first paragraph, or at least reducing it.

- The introduction can be enriched with some information on the thrombotic risk and inflammation of COVID-19 and the potential benefit of statins in this context (e.g

Ferrari et al. 2021. COVID-19 and Thromboinflammation: Is There a Role for Statins? https://www.scielo.br/pdf/clin/v76/1807-5932-clin-76-e2518.pdf).

Second paragraph

- Wouldn't it be "REGN-COV2"?

- I suggest "human immunodeficiency virus (HIV)".

- I suggest "nuclear factor-κB (NF-κB)".

- I suggest "angiotensin converting enzyme-2 (ACE2) receptor".

MATERIALS AND METHODS

- Please add the patient's form of diagnosis (Positive SARS-CoV-2 PCR? Others?)

- Approval of the study: is there an identification number that can be specified?

Exposure and outcomes

- I suggest "serum interleukin-6" and not "serum interleukinS-6".

Line 13

- Only "ACE inhibitors" and not "angiotensin-converting enzyme (ACE) inhibitors", and "ARBs" and not "ARB".

RESULTS

- I suggest adding "years-old" to the values "(64.9 +- 13 vs 45.5 +- 16.6), to make it clear to the reader that they are referring to years of age.

- The authors classify primary outcomes were mortality, defined as prolonged hospital stay and/or need for invasive mechanical ventilation. I miss information regarding the percentage of patients who required the use of mechanical ventilation (statin users vs. statin non-users). The use of statins favored or increased the risk of needing mechanical ventilation? If available, I suggest that they are also in the abstract.

- Are there details about the anti-viral drugs that the patients received?

Risk factors for severe COVID-19 infection

- Only "PSM" and not "propensity-score matching".

DISCUSSION

- Only "PSM" and not "propensity-score matching".

- Please "intensive care unit (ICU)".

- Fifth paragraph: Only "ACE2" and not "angiotensin-converting enzyme 2 (ACE2)".

- Sixth paragraph: Only "PSM" and not "propensity-score matching".

- In the sentence: "The heterogeneous nature of our study, and the large sample size, allowed us to assess the relationship between COVID-19... thus improving your study's generalizability to other hospitalized patients with COVID-19... Additionally, the use of PSM...", i suggest that the authors are not so emphatic. I don't think it's such a large sample of patients using statins, and even using PSM, the study is still retrospective, with its potential biases, even though they are minimized by PSM, as commented by the authors.

Reviewer #3: The manuscript by Samuel K. Ayeh et al investigates the relationship between statin use and COVID-19 mortality and disease severity in patients admitted to the Johns Hopkins health system.

Major Comments

1. The Authors chose the IL-6 level as one of the parameters assessed. What about other inflammatory markers such as CRP and procalcitonin?

2. There is no information on the antiviral treatment (remdesivir?) hospitalized patients.

3. The Authors should provide (or at least discuss in the limitations) value of chest CT quantitative pulmonary inflammation index (PII), passive oxygen therapy, HFNOT therapy, oxygen saturation in hospitalized patients.

4. The strengths and limitations of the study should be deeply addressed, taking into account sources of potential bias or imprecision: Discuss both direction and magnitude of any potential bias.

Minor comments

1. Employment status is unnecessary

2. The Authors should add list of abbreviations

Reviewer #4: Interesting study that showed that statin use is not associated with reduced mortality in those with COVID-19. These results are in contrast to some of the previous findings wherein it was noted that statin use is beneficial to those with COVD-19.

Is it possible that only some types of statins show the beneficial action but not all. Such subgroup analysis may be done and commented upon by the authors. Perhaps, the number of study subjects is not sufficient for such a

subgroup analysis.

Another caveat could be the dose of statins used. Authors may comment on this.

Is it possible that the statins used, their dose and the duration of treatment is not sufficient to reduce plasma levels of IL-6, TNF and other inflammatory cytokines to show their beneficial action. Authors need to comment on this.

Reviewer #5: Statins use and COVID-19 outcomes in hospitalized patients

I thank you for the opportunity to comment this study. The topic of the research is topical.

The conclusion is the following: Statin use was not associated with altered mortality, but with an 18% increased risk of severe COVID-19 infection.

Comments

I want to first highlight how the severe SARS-CoV-2 infection is defined. As an example, I give one example,

For epidemiologic purposes, severe Covid-19 in adults is defined as dyspnea, a respiratory rate of 30 or more breaths per minute, a blood oxygen saturation of 93% or less, a ratio of the partial pressure of arterial oxygen to the fraction of inspired oxygen (Pao2:Fio2) of less than 300 mm Hg, or infiltrates in more than 50% of the lung field (JAMA 2020;323:1239-42).

In this study the severe COVID-19 was defined as follows: prolonged hospital stay (≥ 7 days) and/or need for invasive mechanical ventilation.

Authors should mention in the methodology and in the discussion this issue.

Comparison with other studies

Spiegeleer et al., 2020 (J Am Med Dir Assoc) used the following definition for severe COVID-19: serious COVID-19 defined as long-stay hospital admission or death within 14 days of disease onset.

Daniels et al., 2020 (Am J Cardiol) don’t seem define severe COVID-19 in detailed: “The primary outcome was severe disease, defined as either admission to the ICU or death.

Authors mention in the discussion Rossi et al., 2020, but in this study, patients were not divided according to the severity of the infection. This applies also for the study by Peymanni et al., 2021 (Transl Med Commun).

Authors mention also the meta-analysis by Hariyanto & Kurniawan, 2020 (Diabetes Metab Syndr Clin Res Rev) which possibly seems to support the authors findings. This meta-analysis is uniquely Google Scholar based which is not ensuring that major studies were involved in this analysis. I didn’t find any classification based on severity of COVID-19 in this study. Unfortunately, authors don’t mention f. ex. the larger meta-analysis which has been carried out among hospitalized COVID-19 patients (Kow & Hasan, 2020). In this meta-analysis the findings were different.

It seems that the comparison with other studies is not carried out carefully and objectively. Authors need to be much more specific in comparisons. Currently, there are several studies regarding statin use among hospitalised patients. The current discussion seems to select subjectively those studies supporting the current finding in this study.

Medications

Authors have detailed collected data regarding statin treatment. Why was the dose of statins not collected? There is also data regarding ACE inhibitors. But the date regarding other medications is not collected among these patients. I can’t understand why. There is also no data regarding possible corticosteroid or antiviral medications.

It has been shown in many studies involving hospitalization and statin treatment that statin treatment is discontinued during the hospitalization (Torres-Peña et al. Drugs 2021).

In this well-designed study. It can be calculated that over 60% of hospitalized patients with COVID-19 had for unknown reasons stopped their statin consumption during the hospitalization period. How did the authors monitor statin use in this study?

It is also known that many patients use statins irregularly or even have prescriptions and don’t use them. How was this sorted out? An additional question is how long these patients have used statins?

Mechanisms

Regarding Dashti-Khavidaki & Khalili, 2020 (Pharmacotherapy) they recommend continuing statin use if patient with COVID-19 is hospitalised.

Regarding references [35-37] I understand that they are not related to statins. I can’t see the relevance of them. Please, modify this part of the discussion.

Regarding the ACE2, please, refer to Lei et al. 2021 (SARS-CoV-2 spike protein impairs endothelial function via downregulation of ACE2, Circulation Research) and find the complexity regarding ACE2 upregulation or downregulation.

Conclusion

I think that it is not justified to carry out studies continually in which statins are discontinued among the hospitalised patients. Please adjust this conclusion.

6. PLOS authors have the option to publish the peer review history of their article (what does this mean?). If published, this will include your full peer review and any attached files.

Reviewer #1: **Yes: **Andree Kurniawan

Reviewer #2: **Yes: **Filipe Ferrari

Reviewer #3: No

Reviewer #4: **Yes: **Undurti N Das

Reviewer #5: **Yes: **Alpo Vuorio

---

## [Author Response · Author response to Decision Letter 0]

13 Aug 2021

Reviewer Recommendations and Comments for Manuscript Number PONE-D-21-15364 

Reviewer: #1

The authors have done evaluation from the secondary data regarding the statin use and severe covid-19. My comments are

1. Taking the conclusion in statin use in COVID -19 should be carefull. We should take into account, type, dose statin use, how long the statin have been use before the subject got infection. As we know statin needed time to work the anti inflammmation. should be discussed this issue in the discussion.

Response: We thank the Reviewer for the comment. We have modified the text in the revised Discussion to temper the conclusions, and to emphasize the lack of information on statin type, dose, and duration of therapy prior to COVID-19 infection. 

2. In the results showed that, age, BMI, comorbidity related to the risk of COVID-19. Was the statin use related to severe covid-19 or because any interaction or confounder with comorbidity because they with many co morbidity tend to use statin? should also be discussed.

Response: We appreciate the Reviewer’s comment. As the Reviewer suggests, atherosclerotic cardiovascular disease (ASCVD) is associated with increasing age and BMI. Alhough these co-morbidities may confound outcomes associated with statin use, we took them into consideration in the multi-variable regression models. Nevertheless, we have listed the possibility of additional confounding factors associated with statin use as a limitation of our study in the revised Discussion. 

3. Take the conclusion from the retrospective study should be carefully. May be covid-19 did not improved covid-19 outcome, however further randomized study should confirm this result,

Response: We agree with the Reviewer. As such, in the revised Discussion, we acknowledge the limitations of our retrospective study design, and emphasize the need for randomized trials to address the effect of statins on clinical outcomes following COVID-19 infection.

Reviewer: #2

 First of all, I compliment the authors for their excellent work, which is well-written. In fact, the potential benefit of statins in the COVID-19 scenario is a very important issue, and one that is still controversial.

Below, I respectfully present to the authors my suggestions for improving the manuscript. Please consider the suggestions that you think are relevant.

We appreciate the Reviewer’s positive feedback on our study.

ABSTRACT

- In Methods: Were patients diagnosed with COVID-19 by laboratory tests? I suggest: "A retrospective analysis of patients with laboratory-confirmed COVID-19 admitted to the Johns Hopkins Medical Institutions between..."

Response: We appreciate the Reviewer’s suggestion, and the proposed modification has been made in the revised Abstract (Methods), as well as in the Materials and Methods (“2.1. Study design and participants” section). 

- Please, "risk ratio (RR)".

Response: The abbreviation “RR” has been expanded as “risk ratio” in the “Results” section of the revised Abstract and in the “Risk factors for COVID-19-related mortality” of the Results. 

- In the results, I suggest describing "After propensity-score matching, the average treatment effect..." and/or "after adjusting for age, etc..."

Response: We appreciate the Reviewer’s suggestion. We have modified the text as suggested in the “Results” section of the revised Abstract. 

- In Conclusion, I believe that the authors could draw attention to the importance of using PSM, but that, on the other hand, the data should be viewed with caution, given that the study is retrospective.

Response: We appreciate the suggestion, and have modified the Conclusions of the Abstract accordingly.

BACKGROUND 

- The first paragraph contains information that has already been published extensively (... firts reported to cause severe pneumonia in China's Wuhan... pandemic

status by March 2020...). Thus, I suggest that the authors start by going more directly to the point, excluding the first paragraph, or at least reducing it.

Response: We thank the Reviewer for the suggestion. We have reduced the first paragraph of the revised Background section, as suggested. 

- The introduction can be enriched with some information on the thrombotic risk and inflammation of COVID-19 and the potential benefit of statins in this context (e.g

Ferrari et al. 2021. COVID-19 and Thromboinflammation: Is There a Role for Statins? https://www.scielo.br/pdf/clin/v76/1807-5932-clin-76-e2518.pdf).

Response: We thank the Reviewer for the suggestion. In the revised Background, we have included the potential theoretical benefit of statins in reducing the thrombotic risk associated with COVID-19 and have cited the suggested reference. 

Second paragraph

- Wouldn't it be "REGN-COV2"?

Response: Yes, correct. This change has been made. 

- I suggest "human immunodeficiency virus (HIV)".

Response: We have made this suggested correction.

- I suggest "nuclear factor-κB (NF-κB)".Response: We have made this suggested correction.

- I suggest "angiotensin converting enzyme-2 (ACE2) receptor".

Response: We have made this suggested correction.

MATERIALS AND METHODS

- Please add the patient's form of diagnosis (Positive SARS-CoV-2 PCR? Others?)

Response: We thank the Reviewer for the suggestion and have added this information in the revised Materials and Methods section.

- Approval of the study: is there an identification number that can be specified?

Response: As per our institutional policy, we have excluded this information from the manuscript, but are happy to share it with the Editor/Reviewers: IRB00251829.

Exposure and outcomes

- I suggest "serum interleukin-6" and not "serum interleukinS-6".

Response: We have made these suggested changes.

Line 13

- Only "ACE inhibitors" and not "angiotensin-converting enzyme (ACE) inhibitors", and "ARBs" and not "ARB".

Response: We have made these corrections, as suggested. 

RESULTS

- I suggest adding "years-old" to the values "(64.9 +- 13 vs 45.5 +- 16.6), to make it clear to the reader that they are referring to years of age.

Response: We thank the Reviewer for the suggestion, and have made the requested changes.

- The authors classify primary outcomes were mortality, defined as prolonged hospital stay and/or need for invasive mechanical ventilation. I miss information regarding the percentage of patients who required the use of mechanical ventilation (statin users vs. statin non-users). The use of statins favored or increased the risk of needing mechanical ventilation? If available, I suggest that they are also in the abstract.

Response: We appreciate the Reviewer’s comment. Since the proportion of patients requiring invasive mechanical ventilation was considered as part of the composite outcome (severe COVID-19), we have not reported it in the Abstract or Table 1 as a separate outcome. 

- Are there details about the anti-viral drugs that the patients received?

Response: We appreciate the Reviewers comment. We have included a table containing all the anti-viral drugs that were prescribed to this cohort of hospitalized patients (Supplementary Table S5). 

Risk factors for severe COVID-19 infection

- Only "PSM" and not "propensity-score matching".

Response: We thank the Reviewer for drawing our attention to this repetition. We have substituted “PSM” throughout the revised text after the first instance in which the term/abbreviation has been introduced. 

DISCUSSION

- Only "PSM" and not "propensity-score matching".

Response: The suggested change has been made, as indicated above.

- Please "intensive care unit (ICU)".

Response: The suggested change has been made. 

- Fifth paragraph: Only "ACE2" and not "angiotensin-converting enzyme 2 (ACE2)".

Response: This correction has been made in the revised Discussion.

- Sixth paragraph: Only "PSM" and not "propensity-score matching".

Response: This change has been made, as suggested, in the revised Discussion. 

- In the sentence: "The heterogeneous nature of our study, and the large sample size, allowed us to assess the relationship between COVID-19... thus improving your study's generalizability to other hospitalized patients with COVID-19... Additionally, the use of PSM...", i suggest that the authors are not so emphatic. I don't think it's such a large sample of patients using statins, and even using PSM, the study is still retrospective, with its potential biases, even though they are minimized by PSM, as commented by the authors.

Response: We agree with the Reviewer’s comment. We have revised the text in the Discussion so that it is less emphatic, while highlighting the limitations of the study.

Reviewer #3: The manuscript by Samuel K. Ayeh et al investigates the relationship between statin use and COVID-19 mortality and disease severity in patients admitted to the Johns Hopkins health system.

Major Comments

1. The Authors chose the IL-6 level as one of the parameters assessed. What about other inflammatory markers such as CRP and procalcitonin?

Response: We appreciate the Reviewer’s comment. Given the recent attention on the significant role of IL-6 in activating STAT3 and the NF-κB pathway, resulting in increased production of pro-inflammatory cytokines and chemokines (Huang C et al. 2020 and Rothan HA et al. 2020), we decided to focus on this analyte. We have included a justification in the revised Methods section. 

2. There is no information on the antiviral treatment (remdesivir?) hospitalized patients. 

Response: We thank the Reviewer for the comment. Remdesivir use was not captured in our dataset. However, as described above, we have included a table containing all anti-viral medications administered during hospitalization (Supplementary table S5).

3. The Authors should provide (or at least discuss in the limitations) value of chest CT quantitative pulmonary inflammation index (PII), passive oxygen therapy, HFNOT therapy, oxygen saturation in hospitalized patients. 

Response: We appreciate the Reviewer’s comment. This information was not extracted from the database. In the revised Discussion, we have acknowledged this as a limitation of our study and have listed these parameters as additional potentially useful markers of disease severity. 

The strengths and limitations of the study should be deeply addressed, taking into account sources of potential bias or imprecision: Discuss both direction and magnitude of any potential bias. 

Response: We thank the Reviewer for the suggestion. In the revised manuscript, we have included an in-depth discussion of sources of potential bias, as well as the anticipated direction/magnitude of the bias. 

Minor comments -

1. Employment status is unnecessary

Response: We appreciate the Reviewer’s comment. We have included employment status as a surrogate for socioeconomic status, since earlier publications (e.g., Lassale et al. 2020) found an association between socioeconomic status and COVID-19 outcomes.

2. The Authors should add list of abbreviations

Response: We thank the Reviewer for the suggestion. A complete list of abbreviations have been included in the revised manuscript.

Reviewer #4: Interesting study that showed that statin use is not associated with reduced mortality in those with COVID-19. These results are in contrast to some of the previous findings wherein it was noted that statin use is beneficial to those with COVD-19.

Is it possible that only some types of statins show the beneficial action but not all. Such subgroup analysis may be done and commented upon by the authors. Perhaps, the number of study subjects is not sufficient for such a subgroup analysis.

Response: We appreciate the Reviewer’s comment. Although our findings differed from those of previously published studies, we would like to highlight that they are consistent with those of others (Song et al. 2020, Saeed et al. 2020, and Tan et al. 2020). We agree that a subgroup analysis could potentially shed light on whether COVID-19 outcomes are statin type-specific. However, we were unable to perform such an analysis since over three quarters of our cohort were receiving atorvastatin, and very few patients received any other statin. 

Another caveat could be the dose of statins used. Authors may comment on this.

Response: We appreciate the Reviewer’s comment. We have addressed the issue of statin dose in the revised Discussion.

Is it possible that the statins used, their dose and the duration of treatment is not sufficient to reduce plasma levels of IL-6, TNF and other inflammatory cytokines to show their beneficial action. Authors need to comment on this.

Response: We appreciate the Reviewer’s comment. Multivariable adjustments did not include the inflammatory cytokine, IL-6 when assessing for mortality, but this parameter was included when assessing for severe COVID-19 infection for statistical reasons. The direct relationship between IL-6 and statin use was not explored since it did not make it into the final model. In the revised Discussion, we have addressed the possibility that statin dose/duration of therapy was insufficient to demonstrate a potential beneficial action.

Reviewer #5: Statins use and COVID-19 outcomes in hospitalized patients

I thank you for the opportunity to comment this study. The topic of the research is topical.

The conclusion is the following: Statin use was not associated with altered mortality, but with an 18% increased risk of severe COVID-19 infection.

Comments 

I want to first highlight how the severe SARS-CoV-2 infection is defined. As an example, I give one example,

For epidemiologic purposes, severe Covid-19 in adults is defined as dyspnea, a respiratory rate of 30 or more breaths per minute, a blood oxygen saturation of 93% or less, a ratio of the partial pressure of arterial oxygen to the fraction of inspired oxygen (Pao2:Fio2) of less than 300 mm Hg, or infiltrates in more than 50% of the lung field (JAMA 2020;323:1239-42).

In this study the severe COVID-19 was defined as follows: prolonged hospital stay (≥ 7 days) and/or need for invasive mechanical ventilation.

Authors should mention in the methodology and in the discussion this issue.

Response: We appreciate the Reviewer’s comment and have highlighted how our definition of “severe COVID-19” differs from that of other epidemiological studies in the revised Methods and Discussion sections.

Comparison with other studies

Spiegeleer et al., 2020 (J Am Med Dir Assoc) used the following definition for severe COVID-19: serious COVID-19 defined as long-stay hospital admission or death within 14 days of disease onset.

Daniels et al., 2020 (Am J Cardiol) don’t seem define severe COVID-19 in detailed: “The primary outcome was severe disease, defined as either admission to the ICU or death.

Authors mention in the discussion Rossi et al., 2020, but in this study, patients were not divided according to the severity of the infection. This applies also for the study by Peymanni et al., 2021 (Transl Med Commun).

Authors mention also the meta-analysis by Hariyanto & Kurniawan, 2020 (Diabetes Metab Syndr Clin Res Rev) which possibly seems to support the authors findings. This meta-analysis is uniquely Google Scholar based which is not ensuring that major studies were involved in this analysis. I didn’t find any classification based on severity of COVID-19 in this study. Unfortunately, authors don’t mention f. ex. the larger meta-analysis which has been carried out among hospitalized COVID-19 patients (Kow & Hasan, 2020). In this meta-analysis the findings were different.

Response: We thank the Reviewer for these insights. We agree that there is no widely accepted, standard definition of “severe COVID-19” across studies. Therefore, we have clearly defined the term for the purposes of our study in the revised Methods section, and highlighted how our definition differs from that of other similar studies, as well as potential implications fot study outcomes, in the revised Discussion. Also, we thank the Reviewer for pointing out more recent studies on statin use and COVID-19 outcomes in hospitalized patients, which have been published since our paper was under review. We have cited these new references, including Kow & Hasan, 2020, in the revised manuscript.

It seems that the comparison with other studies is not carried out carefully and objectively. Authors need to be much more specific in comparisons. Currently, there are several studies regarding statin use among hospitalised patients. The current discussion seems to select subjectively those studies supporting the current finding in this study.

Response: We appreciate the Reviewer’s feedback. We have reported the findings from the study by Rodriguez-Nava et al., 2020, which found a mortality benefit with statin use. We have cited additional recently published studies supporting and contradicting our findings in the revised Discussion. 

Medications

Authors have detailed collected data regarding statin treatment. Why was the dose of statins not collected? There is also data regarding ACE inhibitors. But the date regarding other medications is not collected among these patients. I can’t understand why. There is also no data regarding possible corticosteroid or antiviral medications.

It has been shown in many studies involving hospitalization and statin treatment that statin treatment is discontinued during the hospitalization (Torres-Peña et al. Drugs 2021).

In this well-designed study. It can be calculated that over 60% of hospitalized patients with COVID-19 had for unknown reasons stopped their statin consumption during the hospitalization period. How did the authors monitor statin use in this study?

It is also known that many patients use statins irregularly or even have prescriptions and don’t use them. How was this sorted out? An additional question is how long these patients have used statins?

Response: We thank the Reviewer for these insightful comments. In the revised manuscript, we have included a supplementary table containing information on antiviral drugs used (Table S5). Data on steroid use was not included in the study analysis, although we know from clinical experience that during the time period representing the clinical cohort, steroid use was very limited. Contrary to the study of Torres-Pena et al. statin use captured in this study refers to statins administered to patients during the period of their hospitalization, of which > 75% of patients received atorvastatin. Unfortunately, we cannot compare the statin usage pre- and post-hospitalization since prevalence of statin use before hospitalization is not captured by our study. Data about statin usage, dose, adherence, and length of use pre-hospitalization was unavailable to us. These points have been included as study limitations in the revised Discussion. It should be noted that there is no simple dosing conversion among different statins, as there is with opiates and steroids.

Mechanisms

Regarding Dashti-Khavidaki & Khalili, 2020 (Pharmacotherapy) they recommend continuing statin use if patient with COVID-19 is hospitalised.

Response: We thank the Reviewer and have included this in the revised Discussion. It is important to note, however, that this recommendation is based on expert opinion rather than robust empirical evidence. 

Regarding references [35-37] I understand that they are not related to statins. I can’t see the relevance of them. Please, modify this part of the discussion.

Response: We thank the Reviewer for the comment and have removed these references (and related text) from the Discussion. 

Regarding the ACE2, please, refer to Lei et al. 2021 (SARS-CoV-2 spike protein impairs endothelial function via downregulation of ACE2, Circulation Research) and find the complexity regarding ACE2 upregulation or downregulation. 

Response: We appreciate the Reviewer’s comment. We have included this information in the revised Discussion and cited the proper reference. 

Conclusion

I think that it is not justified to carry out studies continually in which statins are discontinued among the hospitalized patients. Please adjust this conclusion.

Response: We appreciate the Reviewer’s comment and have modified the text in the revised conclusion accordingly.

---

## [Editor Report · Decision Letter 1]

18 Aug 2021

Statins use and COVID-19 outcomes in hospitalized patients

PONE-D-21-15364R1

Dear Dr. Karakousis,

We’re pleased to inform you that your manuscript has been judged scientifically suitable for publication and will be formally accepted for publication once it meets all outstanding technical requirements.

Kind regards,

Aleksandar R. Zivkovic

Academic Editor

PLOS ONE

---

## [Editor Report · Acceptance letter]

1 Sep 2021

PONE-D-21-15364R1 

Statins use and COVID-19 outcomes in hospitalized patients 

Dear Dr. Karakousis:

I'm pleased to inform you that your manuscript has been deemed suitable for publication in PLOS ONE. Congratulations! Your manuscript is now with our production department. 

Kind regards, 

on behalf of

Dr. Aleksandar R. Zivkovic 

Academic Editor

PLOS ONE